# Changing the PrEP Narrative: A Call to Action to Increase PrEP Uptake among Women

**Alina Cernasev** [1,*]**, Crystal Walker** [2]**, Drew Armstrong** [3] **and Jay Golden** [4]

1   Department of Clinical Pharmacy and Translational Science, University of Tennessee Health Science Center College of Pharmacy, 301 S. Perimeter Park Dr., Suite 220, Nashville, TN 37211, USA

2   Department of Health Promotion and Disease Prevention, University of Tennessee Health Science Center College of Nursing, 874 Union Avenue, Suite 301, Memphis, TN 38103, USA; cmarti47@uthsc.edu

3   Department of Clinical Pharmacy and Translational Science, University of Tennessee Health Science Center College of Pharmacy, Regional One Health, 877 Jefferson Avenue, Memphis, TN 38103, USA; darmst11@uthsc.edu

4   Walgreens Specialty Pharmacy, 309 22nd Ave. N., Ste. C, Nashville, TN 37203, USA; jon.golden@walgreens.com

*   Correspondence: acernase@uthsc.edu; Tel.: +1-615-253-5600

**Abstract:** Although the incidence of new cases of human immunodeficiency virus (HIV) has decreased in the past decade, in 2018 more than 7000 women with HIV were diagnosed in the United States (US). Globally, per recent reports, 48% of the new HIV infections were among women. There is still no vaccine to prevent HIV transmission. However, pre-exposure prophylaxis (PrEP) was approved in 2012 by the Food and Drug Administration, providing a powerful tool to block HIV infection and help prevent the subsequent development of acquired immunodeficiency syndrome (AIDS). The uptake of PrEP has been slow globally and among the most vulnerable populations in the US, even though the Centers for Disease Control (CDC) recommended its use in high-risk populations, including women. Furthermore, women represent one-quarter of people living with HIV in the US; however, PrEP is underutilized in this group. Thus, it is imperative to make women's voices heard through conducting more research, ensuring sufficient access to PrEP, and enhancing knowledge about PrEP as a viable prevention strategy for women. This article aims to promote women's health by changing the narrative, providing key information on empowering women, and increasing the usage of PrEP.

**Keywords:** PrEP; truvada for PrEP; women; HIV-negative





## 1. Introduction

In the early 1980s, the landscape of infectious diseases changed dramatically and forever upon the identification of a mysterious disease later termed human immunodeficiency virus (HIV). The HIV virus is the pathogen that causes the disease "acquired immunodeficiency syndrome" [1,2]. Initially, scientists worldwide were intrigued and challenged by a new disease with features that did not appear to resemble characteristics of any known pathogens. Four decades later, there is still no vaccine to prevent HIV transmission. However, there is a medication that can be taken to help reduce the risk of HIV infection and is known as pre-exposure prophylaxis (PrEP). Pre-exposure prophylaxis entails taking an antiretroviral medication (emtricitabine/tenofovir disoproxil fumarate or emtricitabine/tenofovir alafenamide) every day to help reduce the risk of contracting HIV [3].

The current World Health Organization guidelines recommend the initiation of PrEP to any individual at high risk of contracting HIV [4]. PrEP is recommended for people at high risk of being infected with HIV from sex or injection drug use, sexually active gay and bisexual men who are HIV negative, and sexually active heterosexual men and

women who are HIV negative [5]. In the same vein, the General Assembly of the United Nations signed in 2016 a "political declaration to end the AIDS epidemic by 2030," which highlighted the development of collaborations among Member States to promote access to HIV prevention services [6].

Although these recommendations were made, the uptake of PrEP has been slow globally and among the most vulnerable populations in the US [7]. Obstacles that have hindered the uptake of PrEP include cost, stigma, lack of awareness, poor PrEP policies in certain countries, and poor access to health services [8]. For example, Calabrese et al., 2020, suggested PrEP therapy to be marketed in an innovative way that is inclusive to all and not focused on patients who may engage in risky behaviors sexually [9]. The new marketing would promote independence for patients instead of stigma for the medication and those who take it [9]. In the same vein, Goparaju et al. 2017, suggested additional education from providers and decreased stigma from society would benefit women who would be more willing to discuss the need for PrEP [10]. Importantly, women are at high risk of contracting HIV through heterosexual sex [11]. As women continue to be at increased risk of contracting HIV, all healthcare professionals must bring awareness about PrEP to women who qualify for this prevention therapy.

This review provides an overview of the current landscape of PrEP use among women, promotes women's health by changing the narrative, and offers recommendations to increase the usage of PrEP in this understudied population. Thus, PrEP education might empower women to take control and protect themselves from an HIV infection without sole reliance on condom usage by their sexual partner.

## 2. Methods

Due to the limited literature on women's usage of PrEP and the Medical Subject Headings (MESH) terminology to encompass both PrEP and women, the research team used the following broad key terms "PrEP" or "Truvada" and "women" over three months. To capture the story accurately, additional key terms were used according to each specific subsection of this manuscript. Since PrEP was approved in 2012 by the FDA, the inclusion criteria included articles published in English after the 2010s and peer-reviewed articles derived from clinical studies. Case reports and case series were evaluated and included based on the inclusion criteria. Although the inclusion criteria included studies published globally, this article focuses on the underutilization of PrEP by women in the US and explores the associated health disparities that this population of women confronts.

## 3. PrEP Approval: What Does This Mean for Women?

Since the 1980s, when HIV/AIDS was declared a pandemic by the World Health Organization (WHO, Geneva, Switzerland), the scientific world has been in a race against this deadly virus to prevent further spread of the disease. In the race to prevent the spread of HIV, there have been several endeavors that have been successful. Safe sex practices, condom use, and limiting sexual partners, are measures to prevent sexual transmission of HIV [12,13]. In addition, there have been several studies highlighting the importance of antiretrovirals in limiting the spread of HIV. Two studies published almost 20 years ago in HIV serodiscordant couples (one partner is HIV positive, while the other is HIV negative) showed a direct correlation between viral load and likelihood of transmitting HIV [14]. The lower the viral load of the HIV positive partner, the lower the risk of HIV transmission to the HIV negative partner. These studies and many more have led to a new initiative knowns as Treatment as Prevention (TasP). While not discussed in detail in this manuscript, TasP is yet another tool in the fight to end the HIV pandemic [14,15]. So while TasP is another method to help prevent HIV infection, there had not previously been a medication that an HIV negative person could take to prevent sexual transmission of HIV. In the mid-2000s, clinical trials were underway to investigate whether the antiretroviral agent Truvada© (TDF/FTC) could be used to provide additional protection from HIV infection [16,17]. The clinical trials were promising, showing that the daily oral medication could protect against

HIV-1 infection among men who have sex with men (MSM), heterosexual men and women, and intravenous drug users. This treatment would come to be known as pre-exposure prophylaxis, or PrEP [17,18].

A decrease of 42% in HIV-1 incidence with daily use of Truvada© was shown in the pre-exposure prophylaxis initiative (iPrex) trial in men who have sex with men (MSM) [19]. This large-scale global clinical trial also demonstrated that Truvada© decreased the risk for HIV infection in transgender women who have sex with men by 44% [19,20]. Cis-gender women are disproportionally affected by HIV via heterosexual contact [11], but results from early studies that targeted this group were unfavorable. Two studies, the tenofovir-based pre-exposure prophylaxis for HIV infection among African Women (VOICE Study) and the pre-exposure prophylaxis for HIV infection among African Women (FEM-PrEP) study, assessed Truvada© vs. placebo and did not find a difference in the rates of HIV infection [18,21]. This lack of efficacy was thought largely due to non-adherence. However, two additional studies, antiretroviral prophylaxis for HIV prevention in heterosexual men and women (Partners PrEP) and antiretroviral pre-exposure prophylaxis for heterosexual HIV transmission in Botswana (TDF2 Study), did find that Truvada© lowered the risk of HIV infection by 75% and 62.2%, respectively [3,17]. In July 2012, the FDA approved Truvada© as PrEP for daily use in people at high risk of contracting HIV [22]. More recently, Descovy© (TAF/FTC) was approved for PrEP in MSM and transgender women from efficacy shown in the DISCOVER study, in which Descovy© was shown to be non-inferior to Truvada© [23]. However, this study did not enroll cis-gender women, and therefore, Descovy© is not approved for use in this population.

## 4. Disparities in Healthcare Access to PrEP among Women

Although men who have sex with men [24] and transgender women [25] are disproportionately affected by HIV in the US and have been targeted for PrEP use, cis-gender women are also at risk. Nearly 20% of all new HIV diagnoses in the US each year are among women [26], yet PrEP use is suboptimal in this population [27]. PrEP has the potential to help end the HIV epidemic, however, when people who need PrEP the most are not accessing and utilizing it, this greatly and negatively impacts HIV prevention. In 2018, only 7% of women who could benefit from PrEP were prescribed PrEP in the US [27]. This is a disheartening reality, considering the impact that HIV has on women, particularly women from minority ethnic backgrounds, such as Black/African American and Latina women [28]. There are many factors that may directly or indirectly impact PrEP access and uptake by women, and these factors should be considered when targeting women for increased PrEP usage.

The highest rates of poverty in the US are found among women, due to gender wage and wealth gaps, segregation into low-paying jobs, and inadequate or inaccessible public social assistance programs [29]. As a result, healthcare insurance may be limited, which may directly affect healthcare access and subsequent PrEP access. One study has even highlighted the impact of poverty and lack of insurance on limited PrEP access [30]. This is of particular importance when targeting low-income women and women of color for improving PrEP access and uptake, as many are less likely to seek preventative services or have routine visits with a healthcare provider [31]. Without insurance, a one-month supply of PrEP is approximately $2200, presenting a substantial barrier for uninsured patients.

Economic factors are major barriers to accessing healthcare, but so are structural concerns such as implicit bias among healthcare providers. This type of bias, defined as having a preference for or an aversion to a person or group of people, typically due to stereotypes, can negatively impact healthcare access and contribute to health disparities [32–34]. This type of bias is a leading contributor to racial and ethnic disparities in women's health in the United States [35]. For example, if a healthcare provider perceives an African American female patient as non-compliant to medications due to a stereotype of non-compliance among the African American community and chooses not to initiate a conversation about PrEP because of this stereotype, this is a missed opportunity for HIV prevention. This is a

classic example of how negative implicit associations can affect thoughts, actions, and subsequently, healthcare outcomes. What is even more concerning about implicit bias is that it can also negatively impact patients' perceptions and behaviors and create uncomfortable environments [36]. Implicit bias may manifest as poor interpersonal skills from the healthcare provider, and as a result, patients lose trust in their healthcare provider and eventually become less engaged in the healthcare system. This lack of engagement results in decreased opportunities for preventive health and management of chronic diseases [37–39].

In a study regarding hesitancy and receptivity to HIV prevention, it was found that properly educated, black college-aged women were highly likely to accept treatment with PrEP [29] This is of importance, as women have indicated a lack of knowledge about PrEP as a reason for not taking PrEP [40–43], and in some cases this has been directly related to their unmet healthcare needs [40]. Women prefer to engage with regular healthcare providers they can trust, and engagement has been cited as an important factor supporting PrEP initiation among women [40]. If women are engaged in healthcare, they may be more likely to be educated about preventive services such as PrEP. This likelihood may be strengthened when actual and perceived instances of implicit bias are reduced, and women are able to create long-lasting patient-provider relationships that are rooted in trust.

## 5. Changing the PrEP Narrative

PrEP is recommended for uninfected people who are at risk of contracting HIV. However, in the current form, "PrEP is underutilized in this group" [44]. The relative invisibility of studies exploring women's perspective on the usage of PrEP prompts new questions that are addressed in this section: How can we change this perspective? How can we increase awareness? How can we empower women to be more engaged in HIV prevention? Consequently, it is imperative to make women's voices heard through conducting more research, ensuring sufficient access to PrEP, and enhancing knowledge about PrEP as viable prevention strategy for women.

Due to unfavorable results from the VOICE and FEM-PrEP studies [18,21], more evidence is needed to inform healthcare providers when making decisions about PrEP initiation in women. In addition, as new medications are approved for PrEP, cis-gender women must be included in clinical trials as they are also at a disproportionate risk for HIV infection and should be targeted for HIV prevention. It is important to note that global studies have reported the usage of "informal PrEP" [45], which may be a potential option for women who are not interested in taking a daily pill. The "informal PrEP" terminology refers to "off-label" PrEP use or "nonprescribed" [45].

Previous studies attributed the usage of "informal PrEP" in the MSM community due to the high cost of the medication [46]. Informal PrEP was obtained through different methods such as online pharmacies, overseas pharmacies where generic PrEP could be purchased at a decreased cost, and medication sharing of Persons Living With HIV/AIDS (PLWHA) who take antiretroviral treatment [45,46]. A study conducted by Charpentie et al. (2014) highlighted how individuals obtained and took PrEP before the French authorities approved the medication for usage [47]. The authors described how an HIV-positive man shared some of his Truvada tablets with his partner before having sexual contact [47]. Australian researchers speculate that another method of obtaining access to PrEP may involve the informal use of prescribed post-exposure prophylaxis (PEP) treatment [48]. Studies conducted in France, The Netherlands, and Australia have indicated informal use of PrEP [46–50], however, data on informal PrEP use are insufficient, and these studies explored "informal PrEP" in MSM. Thus, there is a need for additional studies on the viability of informal PrEP in women and women's perspective about this option. Furthermore, a recent study conducted in the Netherlands showed that generic PrEP facilitated the participants to change the procurement of "informal PrEP" from overseas to local pharmacies in the Netherlands as the prices decreased.

As studies have indicated that women at risk for HIV are not aware of the existence of PrEP as a viable preventive measure [40–43], particular efforts must focus on improving

awareness. Most PrEP campaigns target men who have sex with men. However, in order to change the narrative about PrEP as an option for women, campaigns must start focusing on women. For example, advertisements could depict women who do not know the HIV status of their sexual partner, who infrequently use condoms during sex, or who have had a recent sexually transmitted infection [51]. Increased representation in marketing campaigns will help to spread a broader message about PrEP use and may make women feel that this in an appropriate method of HIV prevention. Although studies and campaigns related to PrEP have targeted transgender women, increased awareness is similarly warranted in this population. A study that explored trans women's views on the facilitators and the obstacles to PrEP acceptability indicated that very few of the interviewed transgender women were aware of PrEP as a method of HIV prevention. This study also showed the importance of using various dissemination channels to ensure that the message is reached in the transgender community [52].

To alleviate financial concerns regarding PrEP, an option for uninsured or under-insured patients is through the manufacturer's patient assistance program, known as Gilead Advancing Access [27]. In order to qualify for this program, the patient's income must fall below 500% of the federal poverty guideline (FPG), with consideration to the cost of living and the number of individuals in the household [26]. With a valid prescription for Truvada or Descovy, pharmacies or healthcare professionals can enroll a patient into the program either online or over the phone, and the enrollee can be approved within minutes, thus expediting PrEP initiation. Women who are concerned with the cost of obtaining PrEP can be reassured that there is funding available, but healthcare professionals must also be aware that this option exists.

## 6. Clinical Implications and Future Directions

Healthcare providers encompass different professions, including dentists, nurses, nurse practitioners, pharmacists, physicians, and physician assistants, all of whom need increased PrEP awareness in order to promote its use among high-risk populations [53]. Important patient factors such as medical mistrust should be considered when increasing PrEP awareness among healthcare providers as an understanding of this phenomenon will aid in a better understanding of some of the unique barriers that some patient populations face with regard to HIV-related behaviors [54,55]. Identifying barriers enables healthcare providers to address those barriers with patients which in turn may support a change in patient behavior [56].

Healthcare providers could also help increase uptake of PrEP by developing and maintaining a trustful relationship with their patients [54] and providing key messages to this vulnerable population about using PrEP as an HIV prevention strategy. Some of these messages should include increasing awareness about women who would be good candidates for PrEP (i.e., women who have had a recent sexually transmitted infection), providing information about the Gilead Advancing Access program for women who need to address financial concerns, and sharing promising insights about the future use of long-acting injectables (i.e., Cabotegravir) under investigation for PrEP to address concerns related to daily oral adherence.

## 7. Conclusions

Pre-exposure prophylaxis, when taken appropriately, is an effective and well-tolerated option to assist in prevention of HIV transmission. This medication has been tested in several patient populations and has shown efficacy across the board. Further research is needed to advance the study and implementation of PrEP in cis-gender women in order to stop the spread of HIV in this vulnerable group. Given that the awareness of PrEP among women is currently low and additional healthcare challenges have emerged following the COVID-19 pandemic, further research is needed to explore the intrinsic motivators for women to initiate PrEP. There have been several studies published aimed at assessing barriers to PrEP initiation in cis-gender women [57–59]. These studies have found

a multitude of factors to consider in this patient population, including perceptions on PrEP effectiveness, approval from loved ones surrounding PrEP initiation, as well as barriers to PrEP initiation. Even with published studies on PreP uptake in cis-gender women, additional research is needed to further target this vulnerable population. In addition, the role of other obstacles such as geographic location, urban versus rural residence, is not yet known, which is essential information needed to design effective interventions in this population.

**Author Contributions:** A.C. and C.W. researched literature and conceived the study. A.C., C.W., and D.A., and J.G. were involved in protocol development. All authors contributed to the writing of the manuscript. All authors have read and agreed to the published version of the manuscript.

**Funding:** This study received no external funding.

**Institutional Review Board Statement:** Not applicable.

**Informed Consent Statement:** Not applicable.

**Data Availability Statement:** Not applicable.

**Acknowledgments:** We acknowledge Kelli Gerth for proofreading the manuscript.

**Conflicts of Interest:** The authors declare that they have no conflict of interest.

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
