# Peer review of "Changing the PrEP Narrative: A Call to Action to Increase PrEP Uptake among Women"

_women, doi:10.3390/women1020011_

Round 1
Reviewer 1 Report
The focus of this review will be of substantial interest to healthcare providers, researchers, policy makers, and funding agencies. However, this reviewer’s enthusiasm is tempered by the relatively cursory nature of the literature review (considering the substantial and growing body of literature on the topic) and by the limited critical discussion on the topic. Yet, overall, this review addresses an important topical question with implications for HIV prevention and epidemic control.
Any revision of the manuscript should address the major and minor comments listed below so as to meet the Journal’s requirement for publication.
MAJOR POINTS:
1) Lines 71-72: “PrEP is a a powerful tool which blocks infection with HIV and helps to prevent the subsequent development of AIDS.”
Comment: Irrespective of whether this article is targeted towards researchers in the field or non-specialist readers, the authors should provide some basic and essential information on PrEP instead of simply stating it as a “powerful tool which blocks infection.” For instance: PrEP is a prevention choice that involves taking an anti-HIV drug on a daily basis to reduce the risk of contracting HIV infection.
2) Lines 81-83: Obstacles that have hindered the uptake of PrEP include cost, stigma, lack of awareness, poor PrEP policies in certain countries, and poor access to health services.
Lines 254-256: In addition, it is not yet known the role of other obstacles such as geographic location, urban versus rural residence, or stigma, which is essential information needed to design effective interventions in this population.
Comment: Consider including some specific and pertinent references on the topic of stigma associated with PrEP. For instance: Calabrese, 2020 (Understanding, Contextualizing, and Addressing PrEP Stigma to Enhance PrEP Implementation).
3) Lines 106-108: Although condom use is one measure to prevent sexual transmission of HIV, no medication previously on the market could function likewise.
Comment: This reviewer supposes that “previously” refers to the early decades of the HIV/AIDS pandemic. The current anti-HIV medications used in antiretroviral therapy (ART) effectively control the viral load in PLHIV, and evidence from multiple studies has shown that there is effectively no risk of individuals with undetectable viral load transmitting HIV to an HIV-negative partner through sex. So, it might be informative to the readers if the authors can discuss whether, and how, an effective ART can shape the success of the PrEP initiative.
4) Line 123: This lack of efficacy was thought largely due to non-adherence.
Comment: This is an important discussion point because adherence is critical for achieving PrEP efficacy. Consider citing additional recent published work. For instance, recent findings from the first observational open-label demonstration project of PrEP among at-risk cisgender women in the US (Blumenthal et al. 2021) revealed that, besides low retention of study participants, PrEP adherence was not sufficient to achieve protective drug concentrations in most study participants.
5) Lines 197-199: It is important to note that global studies have reported the usage of "informal PrEP," also known as “wild PrEP”, which may be a potential option for women who are not interested in taking a daily pill.
Comment: The authors should briefly explain what these interchangeable terms ("informal PrEP," also known as “wild PrEP”) mean.
6) Lines 236-243: Clinical Implications and Future directions
Comment: This section will benefit from additional forward looking discussion points. For instance, the authors should consider discussing, at least in brief, the prospects and value (in the context of the non-adherence obstacle) of long-acting formulations of anti-HIV drugs currently under investigation/development for PrEP (e.g., long-acting injectable HIV inhibitors cabotegravir and lenacapavir).
7) Lines 237-239: Healthcare providers encompass different professions, including dentists, nurses, nurse practitioners, pharmacists, physicians, and physician assistants, all of whom need to be educated on PrEP use in women and how to reduce barriers to access and uptake.
Comment: Consider citing pertinent supporting references. For instance: Nydegger et al. 2021 (A Longitudinal, Qualitative Exploration of Perceived HIV Risk, Healthcare Experiences, & Social Support as Facilitators & Barriers to PrEP Adoption among Black Women)
8) Lines 241-243: Furthermore, healthcare providers could increase uptake of PrEP by developing and maintaining a trustful relationship with their patients.
Comment: Consider citing additional pertinent references. For instance: Tekeste et al. 2018. (Differences in medical mistrust between black and white women: implications for patient–provider communication about PrEP)
9) Lines 246-248: This medication has been tested in several patient populations and has shown efficacy across the board. However, cis-gender women are often excluded from these clinical trials
Comment: If the factor(s) underlying the exclusion of cis-gender women in clinical trials are evident, the authors should consider discussing the same in the main text of the article.
10) Lines 253-254: further research is needed to explore the intrinsic motivators for women to initiate PrEP.
Comment: This is an important discussion point. It will be helpful if the authors briefly reiterate what they mean by “the intrinsic motivators.” Also, consider citing and discussing additional published work. For instance: Teitelman et al. 2020 (Beliefs Associated with Intention to Use PrEP Among Cisgender U.S. Women at Elevated HIV Risk).
MINOR POINTS:
1) Lines 41-42: in 2018 more than 7,000 women were diagnosed in the United States
Suggested revision: in 2018 more than 7,000 women were diagnosed with HIV in the United States
2) Lines 42-43: Globally, per recent reports, 48% of HIV infections were among women.
Suggested revision: Globally, per recent reports, 48% of the new HIV infections were among women.
3) Line 43: There still no vaccine to prevent HIV transmission
Suggested revision: There is still no vaccine to prevent HIV transmission
4) Lines 65-67: identification of a mysterious disease later termed human immunodeficiency virus (HIV), which resulted in the collection of symptoms termed ‘AIDS’-Acquired Immunodeficiency Syndrome.
Suggested revision: Please note that the disease is not termed as “human immunodeficiency virus”. The human immunodeficiency virus is the pathogen that causes the disease “Acquired immunodeficiency syndrome"
5) Lines 70-71: PrEP is a a powerful tool
Suggested revision: PrEP is a a powerful tool
6) Lines 185-186: It is important to note that although women represent one-quarter of people living with HIV (PLWH) in the US, PrEP is underutilized in this group.
Suggested revision: Please rephrase this sentence for clarity. PrEP is recommended for uninfected people who are at risk of contracting HIV. However, in the current form, “PrEP is underutilized in this group” could be misconstrued to mean the “one-quarter of people living with HIV”
7) Lines 204-206: Australian researchers speculate that another method of obtaining access to PrEP may involve the informal use of prescribed post-exposure prophylaxis (PEP) treatment.
Suggested revision: Please rephrase this statement for clarity. It is not clear how the informal use of PEP treatment could constitute “another method” of obtaining access to PrEP.
8) Line 207: informal used of PrEP
Suggested revision: informal use of PrEP
9) Lines 245-246: Pre-exposure Prophylaxis is an effective and well tolerated option to assist in prevention of HIV transmission when taken appropriately.
Suggested revision: Please rephrase this statement for clarity. For instance: Pre-exposure Prophylaxis, when taken appropriately, is an effective and well tolerated option to assist in prevention of HIV transmission.
10) Lines 249-250: Further studies are needed to advance the study and implementation of PrEP in cis-gender women in order to further stop the spread of HIV in this vulnerable group.
Author Response
Reviewer 1
1) Lines 71-72: “PrEP is a a powerful tool which blocks infection with HIV and helps to prevent the subsequent development of AIDS.”
Comment: Irrespective of whether this article is targeted towards researchers in the field or non-specialist readers, the authors should provide some basic and essential information on PrEP instead of simply stating it as a “powerful tool which blocks infection.” For instance: PrEP is a prevention choice that involves taking an anti-HIV drug on a daily basis to reduce the risk of contracting HIV infection.
Thank you for this recommendation. We added more information to the text.
2) Lines 81-83: Obstacles that have hindered the uptake of PrEP include cost, stigma, lack of awareness, poor PrEP policies in certain countries, and poor access to health services.
Thank you for this suggestion. We included another study about stigma to strengthen our argument.
Lines 254-256: In addition, it is not yet known the role of other obstacles such as geographic location, urban versus rural residence, or stigma, which is essential information needed to design effective interventions in this population.
Comment: Consider including some specific and pertinent references on the topic of stigma associated with PrEP. For instance: Calabrese, 2020 (Understanding, Contextualizing, and Addressing PrEP Stigma to Enhance PrEP Implementation).
Thank you for this thoughtful recommendation. This reference strengthens our manuscript.
3) Lines 106-108: Although condom use is one measure to prevent sexual transmission of HIV, no medication previously on the market could function likewise.
Comment: This reviewer supposes that “previously” refers to the early decades of the HIV/AIDS pandemic. The current anti-HIV medications used in antiretroviral therapy (ART) effectively control the viral load in PLHIV, and evidence from multiple studies has shown that there is effectively no risk of individuals with undetectable viral load transmitting HIV to an HIV-negative partner through sex. So, it might be informative to the readers if the authors can discuss whether, and how, an effective ART can shape the success of the PrEP initiative.
We appreciate this suggestion. We made the changes by adding a paragraph about ART. The text reads better, and we thank you.
4) Line 123: This lack of efficacy was thought largely due to non-adherence.
Comment: This is an important discussion point because adherence is critical for achieving PrEP efficacy. Consider citing additional recent published work. For instance, recent findings from the first observational open-label demonstration project of PrEP among at-risk cisgender women in the US (Blumenthal et al. 2021) revealed that, besides low retention of study participants, PrEP adherence was not sufficient to achieve protective drug concentrations in most study participants.
We appreciate and value the recommended studies to include in this manuscript. We amended the text.
5) Lines 197-199: It is important to note that global studies have reported the usage of "informal PrEP," also known as “wild PrEP”, which may be a potential option for women who are not interested in taking a daily pill.
Comment: The authors should briefly explain what these interchangeable terms ("informal PrEP," also known as “wild PrEP”) mean.
We value your recommendation. We amended the text and included a recent study conducted in the Netherlands, making this section clearer.
6) Lines 236-243: Clinical Implications and Future directions
Comment: This section will benefit from additional forward looking discussion points. For instance, the authors should consider discussing, at least in brief, the prospects and value (in the context of the non-adherence obstacle) of long-acting formulations of anti-HIV drugs currently under investigation/development for PrEP (e.g., long-acting injectable HIV inhibitors cabotegravir and lenacapavir).
Thank you for this valuable suggestion. The text reflects the recommendation.
7) Lines 237-239: Healthcare providers encompass different professions, including dentists, nurses, nurse practitioners, pharmacists, physicians, and physician assistants, all of whom need to be educated on PrEP use in women and how to reduce barriers to access and uptake.
Comment: Consider citing pertinent supporting references. For instance: Nydegger et al. 2021 (A Longitudinal, Qualitative Exploration of Perceived HIV Risk, Healthcare Experiences, & Social Support as Facilitators & Barriers to PrEP Adoption among Black Women)
We are grateful for recommending these studies to improve our manuscript.
8) Lines 241-243: Furthermore, healthcare providers could increase uptake of PrEP by developing and maintaining a trustful relationship with their patients.
Comment: Consider citing additional pertinent references. For instance: Tekeste et al. 2018. (Differences in medical mistrust between black and white women: implications for patient–provider communication about PrEP)
We appreciate these recommendations. We amended the text.
9) Lines 246-248: This medication has been tested in several patient populations and has shown efficacy across the board. However, cis-gender women are often excluded from these clinical trials
Comment: If the factor(s) underlying the exclusion of cis-gender women in clinical trials are evident, the authors should consider discussing the same in the main text of the article.
We are grateful for recommending to discuss the cis-gender clinical trials. We amended the text.
10) Lines 253-254: further research is needed to explore the intrinsic motivators for women to initiate PrEP.
Comment: This is an important discussion point. It will be helpful if the authors briefly reiterate what they mean by “the intrinsic motivators.” Also, consider citing and discussing additional published work. For instance: Teitelman et al. 2020 (Beliefs Associated with Intention to Use PrEP Among Cisgender U.S. Women at Elevated HIV Risk).
We are grateful for recommending these studies to improve our manuscript. We cited them.
MINOR POINTS: Thank you for these recommendations. We amended the text as suggested
1) Lines 41-42: in 2018 more than 7,000 women were diagnosed in the United States
Suggested revision: in 2018 more than 7,000 women were diagnosed with HIV in the United States
2) Lines 42-43: Globally, per recent reports, 48% of HIV infections were among women.
Suggested revision: Globally, per recent reports, 48% of the new HIV infections were among women.
3) Line 43: There still no vaccine to prevent HIV transmission
Suggested revision: There is still no vaccine to prevent HIV transmission
4) Lines 65-67: identification of a mysterious disease later termed human immunodeficiency virus (HIV), which resulted in the collection of symptoms termed ‘AIDS’-Acquired Immunodeficiency Syndrome.
Suggested revision: Please note that the disease is not termed as “human immunodeficiency virus”. The human immunodeficiency virus is the pathogen that causes the disease “Acquired immunodeficiency syndrome"
5) Lines 70-71: PrEP is a a powerful tool
Suggested revision: PrEP is a a powerful tool
6) Lines 185-186: It is important to note that although women represent one-quarter of people living with HIV (PLWH) in the US, PrEP is underutilized in this group.
Suggested revision: Please rephrase this sentence for clarity. PrEP is recommended for uninfected people who are at risk of contracting HIV. However, in the current form, “PrEP is underutilized in this group” could be misconstrued to mean the “one-quarter of people living with HIV”
7) Lines 204-206: Australian researchers speculate that another method of obtaining access to PrEP may involve the informal use of prescribed post-exposure prophylaxis (PEP) treatment.
Suggested revision: Please rephrase this statement for clarity. It is not clear how the informal use of PEP treatment could constitute “another method” of obtaining access to PrEP.
8) Line 207: informal used of PrEP
Suggested revision: informal use of PrEP
9) Lines 245-246: Pre-exposure Prophylaxis is an effective and well tolerated option to assist in prevention of HIV transmission when taken appropriately.
Suggested revision: Please rephrase this statement for clarity. For instance: Pre-exposure Prophylaxis, when taken appropriately, is an effective and well tolerated option to assist in prevention of HIV transmission.
10) Lines 249-250: Further studies are needed to advance the study and implementation of PrEP in cis-gender women in order to further stop the spread of HIV in this vulnerable group.
Reviewer 2 Report
Authors discuss about lack of significant use of PreP in cis-women population and argue for conducting studies in these lines and promote use of prep in these populations.
Author Response
Authors discuss about lack of significant use of PreP in cis-women population and argue for conducting studies in these lines and promote use of prep in these populations.
Response: Thank you for this suggestion. In the light of reviewer’s 1 suggestions about cis-women population, we amended the text and used other references.